# Thiosulfate-Cyanide Sulfurtransferase a Mitochondrial Essential Enzyme: From Cell Metabolism to the Biotechnological Applications

**DOI:** 10.3390/ijms23158452

**Published:** 2022-07-30

**Authors:** Silvia Buonvino, Ilaria Arciero, Sonia Melino

**Affiliations:** Department of Chemical Sciences and Technology, University of Rome Tor Vergata, Via della Ricerca Scientifica 1, 00133 Rome, Italy; silvia.buonvino95@gmail.com (S.B.); ilaria-29@libero.it (I.A.)

**Keywords:** rhodanese, hydrogen sulfide, cyanide, redox system, iron-sulfur clusters, alpha-beta domain, biotechnology

## Abstract

Thiosulfate: cyanide sulfurtransferase (TST), also named rhodanese, is an enzyme widely distributed in both prokaryotes and eukaryotes, where it plays a relevant role in mitochondrial function. TST enzyme is involved in several biochemical processes such as: cyanide detoxification, the transport of sulfur and selenium in biologically available forms, the restoration of iron–sulfur clusters, redox system maintenance and the mitochondrial import of 5S rRNA. Recently, the relevance of TST in metabolic diseases, such as diabetes, has been highlighted, opening the way for research on important aspects of sulfur metabolism in diabetes. This review underlines the structural and functional characteristics of TST, describing the physiological role and biomedical and biotechnological applications of this essential enzyme.

## 1. Introduction

The enzyme thiosulfate sulfurtransferase (TST) (EC: 2.8.1.1), originally identified in 1933 by Lang [1], is widely distributed in both prokaryotes and eukaryotes [1,2,3]. Indeed, Lang reported that rat liver contains an enzyme able to convert cyanide to thiocyanate (SCN^−^) in the presence of thiosulfate (S_2_O_3_^2−^), as shown in Figure 1, and this enzyme was named ‘rhodanese’ from the German name for thiocyanate ‘*rhodanid*’. The tst gene encoding the human TST, also named rhodanese, is located on chromosome 22q12.3 [4]. In mammals, TST/rhodanese is localized in the mitochondria [5] and its distribution in the tissues appears to be tissue-specific, with the highest concentration in the liver and significant amounts in the kidneys, adrenals and thyroid glands [6]. In particular, humans have the highest rhodanese activity in the kidneys, which is twice that of the liver, followed by the lungs, brain, stomach and muscles. In plants the TST enzyme is localized in the cytoplasm, in mitochondria and in chloroplast [7,8]. According to the accepted mechanism [9], the TST enzyme mediates the sulfur transfer from thiosulfate (donor) to cyanide (thiophilic acceptor) via a double displacement reaction. First, TST/rhodanese accepts a sulfane sulfur atom from a donor (e.g., thiosulfate), with the formation of a covalent enzyme–sulfur intermediate (E–S) characterized by a persulfide bond at the sulfhydryl group of the reactive cysteine in the active site (Cys 247 in bovine TST); subsequently, the persulfide sulfur is transferred from the enzyme to the cyanide, recovering the native enzyme form [3,10,11]. The estimated apparent K_m_ of the recombinant human TST are 39.5 ± 2.5 mM and 29 ± 4 mM for thiosulfate and cyanide, respectively [12].

Among the sulfur donors of TST, there is mercaptopyruvate (K_m_ = 2.6 mM), a substrate processed also by mercaptopyruvate sulfurtransferase (MST) in cyanide detoxification. There is strong evidence that TST and MST are evolutionarily related because, in addition to being able to interact with the same substrate (cyanide, thiosulfate, mercaptopyruvate) although with distinct kinetics and different affinities, they show a striking similarity in amino acid sequences around the active site (66% of sequence homology between TST and MST in rat liver) [13]. In mammals, the sulfur transfer reactions catalyzed by rhodanese represent a physiological detoxification mechanism because starting with cyanide leads to the formation of thiocyanide, which is excreted renally and 200 times less toxic than cyanide [14]. In fact, the level of rhodanese in different tissues is higher in animals which, by ingesting a greater quantity of cyanide with their diet, need a greater detoxification efficiency. In particular, since cyanide occurs naturally as cyanogenic glycosides in a number of plants (e.g., sorghum, linseed, clovers, grasses, cassava and bamboo [15]), herbivores are more exposed to cyanide through food than carnivores and, subsequently, they have very high levels of TST in their tissues compared to the same tissues of carnivores [16,17]. TST expression and activity in both the liver and kidneys of pandas were found to be significantly higher than in other animals such as cats. The detoxification of cyanide was suggested as a primary activity of TST also in the case of plants, in which cyanide is endogenously produced especially during the biosynthesis of ethylene in the ripening of fruits and also in the senescence processes of leaves [18].

In the last twenty years, several studies have highlighted the involvement of TST in metabolic processes, as well as its relevance in various metabolic and non-metabolic diseases. This review underlines the structural aspects of this enzyme, which has represented and still represents an enzymatic structural model. Moreover, it highlights the involvement of this mitochondrial enzyme in different and relevant biochemical processes, its involvement in the onset of some diseases, as well as its possible application in biotechnology.

## 2. Structure and Function of TST

### 2.1. TST/Rhodanese Structure: A Model for Studying the Protein Folding

The TST superfamily members are characterized by an alpha-beta topology with a structural module in which alpha-helices surround a central five-stranded beta-sheet core [19]. The sulfurtransferase family members differ for the structural organization of the rhodanese/TST domains [20]. Some TST members are characterized by a single rhodanese domain, such as the bacterial GlpE [21] and PspE [22,23], and human TSTD1 [24], while in other sulfurtransferase proteins, such as TST from humans (hTST), *Bos taurus* (Rhobov) and *Azotobacter vinelandii* (RhdA) [20,25,26], the rhodanese domain is present in a tandem repeat and the active site is located in a cleft between the N- and C-terminal domains [12,27]. TST proteins such as GlpE, ThiI and RhdA exhibit a preferential sulfur transfer activity with thiosulfate as a donor [28,29,30], although RhdA can have other substrates [31,32], while TSTD1 probably plays a role in sulfide signaling, where the persulfidated thioredoxin serves as a donor to TSTD1 and the thiol form of TSTD1 serves as a sulfur acceptor [33]. Figure 2 shows the tridimensional (3D) structure of TSTs. Moreover, there are other proteins, such as in Cdc25 phosphatase [20] or in the PRF and CstB proteins [34,35], that have rhodanese/TST domains without showing sulfurtransferase activity. The best-characterized TST is the Rhobov, which represents the reference structure for the TST/rhodanese subfamily [36] and has 90% similarity in the aminoacidic sequence with hTST. This enzyme represents a structural model of a protein with alpha/beta topology and in several studies it was used in order to clarify the chaperon mechanism of protein folding. Rhodanese has been the classic protein substrate of the remarkable molecular machine GroEL/GroES that binds unfolded proteins and allows them to fold within a cavity formed by the heptameric rings of GroEL/GroES and ATP [37,38,39,40,41]. Rhobov can be also partially folded in the GroEL-GroES-ADP complex, but in that case the protein doesn’t show TST activity [42]. The N-terminal signal sequence (1–23) of Rhobov is crucial for the initial steps of the folding process and, in particular, the first 40 amino acids of the bovine enzyme are essential in the interaction with Gro-EL [43,44]. The crystal structure of both TST isoforms consists of two similar globular structural domains [45] that have rather poor sequence similarity [40,46]. These domains are connected by a tether region and are associated by strong hydrophobic interactions. The slightly smaller C-terminal domain hosts the active site, which is located near the interdomain surface [12], Cys-247 [9] is the active residue directly involved in the enzymatic reaction, and large conformational changes occur during the catalysis due to the structural flexibility of the tether region [47,48]. Protein intrinsic fluorescence was also used to follow the conversion from ES to E [49]. The observed quenching of the intrinsic fluorescence of TST upon the binding of sulfur was also explained as a possible generalized conformational change in the enzyme induced by persulfide formation [50,51,52]. Water proton NMR relaxation studies [53] and ^35^Cl NMR relaxation studies [54] on eukaryotic Rhobov report that there are significant changes in the exposure to solvent or to anion binding for the two catalytic states ES and E. These results have been interpreted as due to important interdomain reorientation(s) between the two structural domains of the enzyme upon the catalytic cycle [54]. However, ^15^N NMR relaxation studies together with essential dynamics studies on the prokaryotic TST from *Azotobacter vinelandii* (RhdA) [55,56] did not show large differences between the two forms indicating that only small conformational rearrangements, probably confined around the active site, occur between the ES and E form. However, all the structural studies on both eukaryotic and prokaryotic TST proteins with double domains are in agreement with an enhanced solvent accessibility in the E form. The presence of the extra sulfur confers a major thermodynamical stability to the ES form of bovine TST (Rhobov) that results in its being ~8 kcal/mol more stable than the E form [57]. The substrate specificity is conferred from the residues on the active-site loop including the cysteine, which is persulfured during the catalysis. In particular, the positive charges of the CRKGVT motif in hTST interact with the negative charges on the substrate (e.g., oxygen atoms in the case of thiosulfate). The double mutants Arg248Gly and Lys249Ser of hTST, to mimic the MST active site, increase to about 17-fold the K_m_ value for thiosulfate and decrease the *k*_cat_ for rhodanese 6-fold [13].

The high number of members of the homology superfamily of rhodanese (accession number: PF00581 (accessed on 1 April 2021); http://sanger.ac.uk/cgi-bin/Pfam) suggests that the rhodanese domain can be involved in distinct physiological roles [20,59].

### 2.2. Functional Role of TST in Cell Metabolism

TST/Rhodanese, in addition to cyanide detoxification, either alone or in association with other proteins, can perform a variety of biological roles, ranging from the transport of sulfur and selenium in biologically available forms [29,60,61] and the mitochondrial import of 5S rRNA [62] to the detoxification processes [48] and restoration of iron–sulfur clusters in Fe-S proteins, such as the aconitase [63] and mitochondrial respiratory complexes [64]. All of these functions are summarized in the Figure 3. The persulfide intermediate (ES form) of the enzyme acts as a sulfur-carrier and plays a critical role in sulfur trafficking by delivering sulfur in a “safe” chemical species in several biosynthetic pathways [65,66,67]. TST is able to interact with enzymes of oxidative metabolism, such as succinate dehydrogenase [64], NADH dehydrogenase [68], xanthine oxidase [69] and NADH nitrate reductase [8]. The iron-sulfur clusters of ferrodoxins, succinate dehydrogenase, and mitochondrial NADH dehydrogenase can be reconstituted by incubation with TST, a sulfur donor and an iron source [23,64,68,70]. Due to these interactions with the enzymes of the ETC (electron transfer chain), TST could have direct control of mitochondrial respiratory activity. Besides, a TST down regulation has been related to a reduced availability of iron-sulfur centers and the rate of electron transport with an increase in superoxide anions formation [71]. Many studies suggest, in fact, that the proteins have roles in ‘managing’ stress tolerance and in maintaining redox homeostasis [63,72,73]. TST is also able to degrade reactive oxygen species (ROS) with thioredoxin in cell-free systems [74,75]. Analyses of TST expression and activity also revealed that it was overexpressed in MST-knockout mice in order to compensate for the effect of the MST deficiency [76].

The E form of TST can be phosphorylated as a result of cellular signaling [77], at serine 124 in bovine TST, which is accessible only in unsulfurated TST. The conformational change in the enzyme induced by the phosphorylation brings the side chain containing Cys-247 into proximity with either Cys-254 or Cys-263 [78]. These cysteines can form a disulfide bridge and render the phosphorylated TST unable to metabolize sulfane sulfur donors. On the contrary, dephosphorylated TST could increase the rate of electron transport and ATP production by catalysis of the reverse reaction and the mobilization of sulfur for iron–sulfur cluster synthesis or repair [79]. TST phosphorylation could be a highly dynamic post-transcriptional process with a steady-state level of modification of iron–sulfur centers in the electron transport chain. TST phosphorylation represents a mechanism by which mitochondria adjust the rate of oxidative metabolism in response to energetic demand. Adaptive changes in the activity and expression of cystathionine-lyase (CSE), MST and TST in various frog tissues in response to exposure to lead, mercury and cadmium confirmed the protective function of these enzymatic proteins against electrophilic stress [80]. TST activity is inhibited by Ca^2+^, Zn^2+^ [3], Cu^2+^ [81] ions, SO3^2−^ [3], SO4^2−^, oxaloacetate, pyruvate [82], H_2_O_2_ [83] and in general by oxidative stress. On the other hand, TST is activated by glutathione [84] and L-cysteine [85], and reducing conditions can reactivate the enzyme. Other activating molecules, such as butyrate and histone-deacetylase inhibitors, have been found to increase TST activity. A significant increase in TST activity and expression was observed in human cancer cell line HT-29 in advanced colon cancer [86], where the expression of both thiosulfate sulfurtransferase and mercaptopyruvate sulfurtransferase was markedly reduced. The gene expression analysis in colonic mucosa from cancerous and normal tissues showed that the tst gene was one out of three mitochondrial genes that had a statistically significant decrease in expression from normal tissue to tumor at every Dukes’ stage A–D, hypothesizing that a possible cause of colorectal cancer carcinogenesis might be located in the mitochondria [87,88].

Another role of TST is in selenium metabolism, as demonstrated in vitro that the E form of TST binds selenium at a 1:1 molar ratio, leading to the formation of the stable perselenide form of TST [32]. Accordingly, TST can have a critical role in the generation of the reactive form of selenium for the synthesis of selenophosphate (SePO_3_), which is an active selenium donor required for SeCys-tRNA synthesis, a relevant precursor of selenocysteine. Therefore, the TST enzyme can have a relevant role in the synthesis of selenoenzymes and so indirectly facilitate the removal of hydrogen peroxide by GSH and provide reducing equivalents to thioredoxin (TRX). All members of the sulfurtransferase family can oxidize and reduce, respectively, the thiol and persulfidated forms of TRX [33,73,89]. In mammalian cells, an interaction of TST with TRX plays a relevant role in the balance of metabolism. TST activity is important for mitochondrial oxidation and could be an effective marker of mitochondrial dysfunction in response to oxidative stress. An organo-sulfane sulfur compound, such as sodium 2-propenyl thiosulfate, was found to induce the inhibition of TST activity in tumor cells. The activity of the enzyme was restored by thioredoxin 2 (TRX2) in a concentration- and time-dependent manner [89]. The TST detoxification function from intra-mitochondrial oxygen free radicals seems also to be protective against persistent oxidative stress, including that induced by radiation, which leads to TST induction [90,91,92,93]. TST, in association with MST, can promote an anti-oxidative stress function by regulation of sulfane sulfur, GSH or thioredoxin leading to the inhibition of the persistent oxidative stress [94]. In agreement with this relevant function of TST, low TST expression has been detected as a negative prognostic marker in hemodialysis patients [71]. During the last decade, a relevant role of TST in the mitochondrial pathway of the endogenous gasotransmitter hydrogen sulfide (H_2_S) has been recognized. However, the detailed molecular role of TST is still debated. TST/Rhodanese is a crucial enzyme in the H_2_S oxidation route that leads to the formation of hydropersulfides (-SSH), thiosulfate and sulfate. TST/Rhodanese catalyzes the transfer of sulfane sulfur from glutathione persulfide (GSSH) to sulfite, which is produced in a reaction catalyzed by persulfide dioxygenase (SDO), i.e., ETHE1, to form thiosulfate [12,95,96].

TST is also able to catalyze the formation of H_2_S using dihydrolipoic acid (DHLA) [97,98] which is also involved in the generation and transport of sulfane sulfur as well as H_2_S production from L- and D- cysteine [46,98]. The ability of TST to catalyze the production of the H_2_S [26,76] has suggested the possibility that rhodanese might be a source of H_2_S in vivo. However, although the reaction is reversible, the efficiency for thiosulfate production is much greater than that for thiosulfate utilization in this reaction [12].

This is suggested due to the activity of the sulfide quinone oxidoreductase (SQR), which is the first enzyme in the mitochondrial sulfide oxidation pathway, and predominantly catalyzes the synthesis of GSSH. The amount of rhodanese is about 4-fold lower than of CSE, but is 14-fold higher than of cystathionine-synthase (CBS) [99], the sulfide pathway enzymes that synthetize H_2_S [100,101]. The ability of TST from murine liver lysate to produce H_2_S from thiosulfate in the presence of GSH is only ~1% of the ability of cystathionine -synthase and -cystathionine lyase to produce H_2_S from cysteine and homocysteine. This suggests a primary role of rhodanese in H_2_S catabolism rather than in its production. The involvement of TST in the metabolism expands the relevance of this enzyme in several biochemical pathways and diseases, considering the remarkable properties of the endogenous H_2_S as a gasotransmitter, cytoprotective agent, mediator of vascular response and platelet adhesion, and regulator of glucose metabolism, redox balance and the detoxification of intra-mitochondrial oxygen free radicals [102,103,104,105,106]. In the following paragraph the effects of the dysregulation of TST expression and activity on the development of some diseases will be discussed.

## 3. TST in Diseases

### 3.1. Lebers Hereditary Optic Neuropathy

Historically, the first disease related to an alteration in the expression of TST/rhodanese is Lebers hereditary optic neuropathy (LHON). LHON is a rare neurodegenerative disease that results in significant visual loss or even total blindness. In most cases, this disease occurs in young men and leads to a selective degeneration of retinal ganglion cells (RGCs) with optic atrophy within a year of disease onset. LHON is caused by the presence of mitochondrial DNA point mutations [107] that affect complex I subunit genes. The dysfunction of NADH: ubiquinone oxidoreductase due to LHON mutations leads to the overproduction of ROS which, in turn, triggers a series of mitochondrial dysfunctions that eventually result in the death of retinal ganglion cells via apoptosis [108,109]. A pronounced reduction in TST activity was found in liver biopsies [110] from Leber patients suggesting a role of TST in this disease. Moreover, the deficiency of the cyanide detoxification pathway plays a role in disease progression. TST activity in rectal mucosa has also been found to be six-fold lower in patients with LHON, and these evidences have suggested that TST deficiency can either cause LHON or be directly involved in its development [111]. Conversely, others [112,113] failed to show evidence of quantitative or qualitative defects on the liver TST isozyme patterns of subjects with LHON. However, there is vast heterogeneity of the samples due to the high tissue-specificity of TST expression and the population-specificity of the mtDNA LHON-associated mutations distribution [114,115,116]. Thus, the differences may be also due to the selection of patients from different ethnic groups and the use of different tissue in the experiments.

### 3.2. Colonic Diseases and Cancer

The effects of alteration in TST expression and activity are detected in several other diseases with mitochondrial dysfunction and/or unbalance in redox state or in H_2_S metabolism, such as colonic diseases and, according to recent studies, also obesity-associated type 2 diabetes mellitus (summarized in Figure 4). TST seems to play a significant role in colonic diseases where it may function primarily by decreasing H_2_S in rectal mucosa. Hydrogen sulfide is normally present in the colonic lumen because of sulfate-reducing bacteria activities and in patients with active ulcerative colitis (UC) H_2_S concentration is increased [117], with a subsequent cytotoxicity effect of this molecule. Accordingly, rhodanese is likely to be the main responsible enzyme for the detoxification of H_2_S in the colon [118]. A reduction in TST activity correlates to the development of ulcerative colitis, conversely mucosal healing is associated with an increased TST gene expression [119]. However, it is not clear if this deregulation in TST level is a primary defect or the result of local inflammation. Besides its role in UC, rhodanese is a protein related to the process of aging [120]. It is decreased in colonic epithelial tissues from old people and may be associated with a slow decline in physiological vigor and an increasing susceptibility to age-related diseases, including colon cancer. Some experimental studies in neoplastic cell lines have highlighted a decrease in the intracellular sulfurtransferase activity, such as TST activity, and of the presence of sulfane sulfur-containing compounds [121,122,123,124]. Recently, TST activity was detected two-fold lower in mouse mammary gland tumor cells (4T1 cell line) as compared to the activity in mouse mammary gland cells (MNuMG cell line) [124]. All this seems to suggest a different sulfur metabolism in the cancer cells with the possibility to develop selective anti-tumor therapies taking into account the differences in the sulfur metabolism with the normal cells.

### 3.3. Diabetes Mellitus Type 2 and Friedreich’s Ataxia

In recent years, a dramatic increase in obesity and in diabetes mellitus type 2 (DMT2) was observed worldwide, due mainly to the globalization of a diet with a high content of animal fat and protein, refined grains and added sugar. Obesity is related not only to an obesogenic environment, but also to a genetic susceptibility [125,126] and similarly DMT2 is linked to the combination of an unhealthy lifestyle (overnutrition and sedentary life) with the genotype of the subject [127,128]. Therefore, for the development of new interventions in order to prevent or reverse excess weight gain and diseases associated with it, it is crucial to clarify the genetic variants that play a major role in inducing susceptibility or resistance to obesity and to DMT2. Recently, the *tst* gene has been identified as a candidate obesity resistance gene involved in low adiposity and metabolic health. Morton et al., using a polygenic lean mouse model, showed a connection between the up-regulation of the tst gene expression and reduced adiposity and insulin sensitization in white adipose tissue [129]. TST enzyme up-regulation, besides the demonstrated antidiabetic and leanness effects, was found to reduce the inflammatory status feature of obese subjects [130]. Furthermore, it was observed that when TST is pharmacologically activated by its substrates, sodium thiosulfate and the garlic compound diallyl disulfide (DADS), this leads to a significant reduction in inflammatory cytokines production in a mouse adipocyte cell model [130]. Among the roles of the tst gene, the quenching of mitochondrial ROS and H_2_S, both involved in adipocyte function, has been mentioned [131,132], leading to maintenance of insulin sensitization in adipose tissue, lipolytic responsiveness and adiponectin release that ultimately drives peripheral oxidative disposal of excess fat. Chronically high levels of ROS cause an elevated oxidative stress in obese individuals [133] and lead to the development of insulin resistance [134,135]. This unbalanced redox state in adipocytes is linked to diverse factors such as the action of circulating free fatty acids highly concentrated in the plasma of obese patients. Among others, palmitate causes the increase in FABP4/aP2 (fatty acid-binding protein 4) expression [136], which results in reduced expression of uncoupling protein 2 (UCP2) [137], determining an increased ROS production. Interestingly, TST-knockdown adipocytes engendered higher levels of mitochondrial ROS after cells were exposed to oxidative stress, further suggesting a functional role of TST on ROS. Moreover, the inhibition of TST activity with 2-propenyl thiosulfate leads to a reduction of ROS-sensitive adiponectin release from a preadipocyte cell line model (3T3-L1). In cells from TST-knockout mice was found a reduced basal lipolysis, a process dose-dependent from the sulfide concentration in the preadipocyte cell line; these findings further support the role of TST in the modulation of H_2_S effects on adipocyte function. Notably, since several recent studies highlight a link between insulin resistance and an altered electron transport chain [138,139] due to the ability of TST to influence mitochondrial respiration, there may be an implication of TST interaction with the iron-sulfur centers, correlated with the anti-diabetic effect exerted by the enzyme. Negative correlation between TST and body mass index and positive correlation with levels of glucose transporter type 4, insulin receptor substrate 1 and peroxisome proliferator activated receptor gamma were also observed [129,140].

Deregulation of tst/rhodanese gene expression has also been related to other diseases, such as Friedreich’s ataxia (FRDA). FRDA is an autosomal-recessive neuro- and cardio-degenerative disease characterized by cardiomyopathy and progressive ataxia is related to the inhibition of frataxin expression [141], a protein involved in iron–sulfur clusters (ISCs) biogenesis. A down-regulation of both serine hydroxymethyltransferase and rhodanese was observed in the fibroblasts and lymphoblast cells from the FRDA patients and the neural NT2 cells line in which the expression of frataxin was inhibited [142].

All these findings that show the association between the expression of rhodanese and degenerative states can make rhodanese a potential tumor/disease biomarker and treatment target.

## 4. Biotechnological Application of TST

In the last few decades, TST, due to the capability of detoxifying cyanide, has been investigated for its use in many environmental and biomedical biotechnologies (see Table 1). Among the most important ones, in this review we emphasize the removal and biodegradation of cyanide from different environmental compartments [143,144], the production of carrier-based therapeutic catalytic bio-scavengers against cyanide [145,146], the reduction of toxic HCN concentration in ruminant animals’ diet [147] and the development of innovative biosensors for cyanide detection [148].

Cyanide is a nitrile which is extensively employed in many industry fields such as pharmaceuticals, food processing, coal cooking, jewelry, mining, electroplating, plastics, dyes and paints [143]. It is present in various forms in industrial wastes (including free cyanide, simple inorganic salts, metal cyanide complexes, cyanate and organic cyanides) [149], in agricultural wastes (nitrile herbicides as dichlobenil, ioxynil or bromoxynil) [150] and in wastes from precious metal mining (roughly 1.0 kg of sodium cyanide or potassium cyanide is needed to recover 1.5 g of gold) [151]. Because of its potential environmental toxicity, different cyanide detoxification strategies, both physical-chemical [152,153] and natural-biological ones [149,154,155,156], have been developed and modified during the last decades for curbing down cyanide pollution. The current physical-chemical approach, often relatively expensive, mostly involves ozonization, chlorination, etc. [157] leading to the release of further toxic reaction by-products in water-bodies, the air and soil.

Instead, the biological methods, which involve the use of biological agents such as microorganisms (bacteria/fungi) and enzymes, represent an inexpensive, eco-friendly, highly efficient approach with no release of toxic by-products [144,152]. Hence, bacteria such as *Escherichia coli* expressing recombinant TST from *Pseudomonas aeruginosa* [158], *Azotobacter vinelandii* [30,159], *Bacillus pumilus* [160] and the TST enzyme isolated from bacteria [23], have been used in the biodegradation of cyanide. Although to a much lesser extent, the use of TST from fungi such as *Rhizopus oryzae* [161], *Trametes sanguinea* [162] and *Aureobasidium pullulans* [144] was also investigated. TST immobilization on supports such as polyacrylamide gels [163] and sepharose [164] has been used extensively, since immobilized enzymes show unique properties with respect to the free enzymes, such as more stability and higher potential for reuse. Ademakinwa et al. produced TST from the yeast-like fungus *Aureobasidium pullulans* and performed an immobilization via cross-linking with glutaraldehyde prior to enzyme entrapment in alginate supports, with the aim to develop a new bio-device for the biodegradation of cyanide in cassava mill effluents [144].

Other applications of TST are more related to civilian and military research, trying to develop effective catalytic bio-scavengers for a therapeutic and prophylactic defense against cyanide as a chemical toxic agent. The treatment of cyanide intoxication requires the use of scavengers (e.g., methemoglobin former sodium nitrite or cobalt compounds or cyanohydrin formers) and/or the transformation of cyanide into the less toxic thiocyanate (SCN^−^) by means of the exogenously administered sulfane sulfur and sulfurtransferase enzymes [165,166,167]. Towards this aim, isolated, purified, recombinant TST from bacteria has been tested in in vitro and in vivo experiments. Frankenberg and colleagues widely investigated the possibility of directly injecting free rhodanese in mice after cyanide poisoning [168]. However, the enzyme activity in plasma decreases rapidly after the intravenous injection mainly due to renal excretion [168]. Therefore, the fast degradation and excretion, together with an adverse immunologic reaction against the free administered enzyme, has limited its use in vivo. To overcome these limitations, in the last decades, the realization of biodegradable, nontoxic TST-carrier systems, microparticles and nanoparticles, characterized by better pharmacokinetic parameters, has been investigated [145,169,170,171]. For example, microparticles (micrometers in diameter), the Carrier Red Blood Cells (CRBCs), and resealed and annealed erythrocytes, have been successfully used as a cyanide antagonist [172,173,174,175]. However, some practical difficulties related to the use of CRBCs such as prior blood typing requirements, the cells’ fragility and the need for sophisticated encapsulation techniques, have pushed towards the use of a different approach, consisting of encapsulating TST within a bioprotective environment such as sterically stabilized long-circulating liposomes (SL) (100–150 nm diameter) [176,177]. Nowadays, dendritic polymers (DPs), of some nanometer size, provide an excellent system for capturing enzymes. Petrikovics et al. focused on nano-intercalated rhodanese using a DP based on hyperbranched poly(2-ethyloxazoline) with a CH_5_(CH_2_)_17_ chain modified surface in combination with the classic inorganic sodium thiosulfate (TS) and a chemical component of garlic, diallyldisulfide (DADS), a sulfur donor, to prevent cyanide lethality in a prophylactic mice model. They showed that DADS is not a more efficient sulfur donor than TS; however, the use of external TST significantly enhanced the in vivo efficacy of both sulfur donor-nitrite combinations, indicating the potential usefulness of enzyme nano-delivery systems in developing antidotal therapeutic agents [178]. Recently, an alternative approach was proposed by Bellini et al. which produced a photo-polymerizable hybrid hydrogel using exopolymeric substances (REPS) extracted by the cyanobacterium *Trichormus variabilis* VRUC 168, with the addition of polyethylene glycol diacrylated (PEGDa) [146]. Bellini and colleagues used the recombinant TST from *Azotobacter vinelandii* [26,30,56,89] as an enzymatic model to assess the enzyme carrier ability of REPS-hydrogel. This enzyme is characterized by the presence of only one Cys residue, which is also the catalytic residue present in the active site and the enzymatic activity is easily evaluated using the Sörbo assay [179]. They demonstrated that the REPS solution did not significantly affect TST activity, even after many hours of incubation and even after photo-polymerization, the hydrogel embedding TST (TSTREPS-Hy) showed enzymatic activity; these findings suggest the possibility to use the REPS-Hy as a TST carrier system for cyanide detoxification [146].

Another potential application of TST concerns the use of this enzyme to reduce ruminant animals’ mortality due to cyanide-containing feed ingestion [147]. In fact, phytotoxins called cyanogenic glycosides are found in many plant species commonly used as a source of energy for ruminant animals in tropical regions [180,181], such as cassava root; these compounds produce highly poisonous HCN when consumed leading to high rate of animals’ mortality, representing a big issue particularly for breeding cattle. Supapong and Cherdthong found that increasing the dose of rhodanese up to 1.0 mg/10^4^ ppm KCN significantly increased the rate of ruminal HCN degradation. Moreover, they found that, though the in vitro dry matter digestibility (IVDMD) was suppressed when increasing doses of KCN were administered at 600 ppm, the supplementation of rhodanese enzymes at 1.0–1.35 mg/10^4^ ppm KCN enhanced IVDMD [147].

Finally, the cyanide-detoxifying ability of rhodanese has also been exploited to develop biosensors for cyanide detection [182,183,184,185]. A fundamental issue for optimal performance in biosensors is preserving the catalytic activity of the enzyme immobilized in bioelectronic devices [186,187]. De Araujo and colleagues have adsorbed rhodanese onto Langmuir–Blodgett (LB) monolayers of the phospholipid di-myristoyl-phosphatidic acid (DMPA), and characterized the enzyme-films by fluorescence spectroscopy, polarization-modulated infrared reflection-absorption spectroscopy (PM-IRRAS), and atomic force microscopy. LB films could be a suitable and innovative strategy for the realization of nanostructured films since monomolecular films can be deposited on solid supports allowing the control of chemical composition and surface density [148]. Accordingly, de Araújo and Caseli showed that the enzyme activity of rhodanese could be better preserved when immobilized as DMPA LB film with respect to the homogeneous medium due to interactions of the polypeptide structure with the phospholipid, resulting in systems with a higher stability after one month. The Rh-DMPA LB film was tested to detect cyanide in a proof-of-concept approach, which allows the use of the Langmuir–Blodgett methodology for the future development of stable colorimetric cyanide sensors with control over the molecular structure [148].

More recently, the application field of TST is represented by the optimization of hydrogel-based cellular scaffolds for tissue engineering and regenerative medicine [146,188,189]. TST was also used in the context of the study and design of a new hydrogel-based scaffold for three-dimensional (3D) cell cultures with potential applications in tissue engineering and regenerative medicine. Mauretti and colleagues produced bovine serum albumin microbubbles (MBs) coated with recombinant TST from *Azotobacter vinelandii* and incorporated the TSTMBs into photo-polymerizable polyethylene glycol-fibrinogen hydrogel (PFHy), with the aim to realize H_2_S-releasing scaffolds to promote the human Lin^−^ Sca-1^pos^ cardiac progenitor cells (hCPCs) proliferation and differentiation [188]. In particular, the proliferation of hCPCs into 3D TSTMBs-PFHy molds was monitored in a cell culture medium with the addition of 3 mM of thiosulfate and a significant increase in cell proliferation with respect to PFHy samples was detected [188]. The observed increase in cell proliferation, obtained both with the addition of TST and thiosulfate in the medium and embedding TSTMBs in the PFHy, was linked to the slow H_2_S release over time which exerts an antioxidant protective effect on cells, therefore stimulating cell growth [188]. All of these biotechnological applications are possible for the great structural and functional stability of this enzyme.

Certainly, this property of TST together with the recent discovery of its relevance in metabolic diseases, such as diabetes, could favor the development of new bio-medical applications of this enzyme in the future.

## Figures and Tables

**Figure 1 ijms-23-08452-f001:**
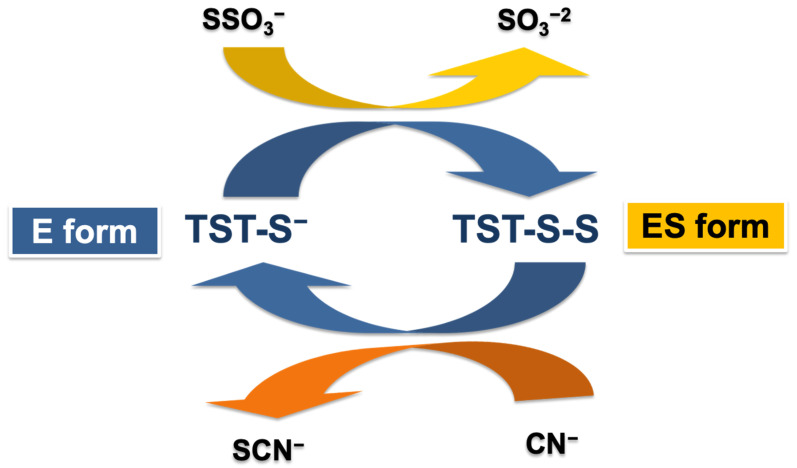
Scheme of the ping-pong mechanism of the sulfur transfer reaction catalyzed by TST/rhodanese.

**Figure 2 ijms-23-08452-f002:**
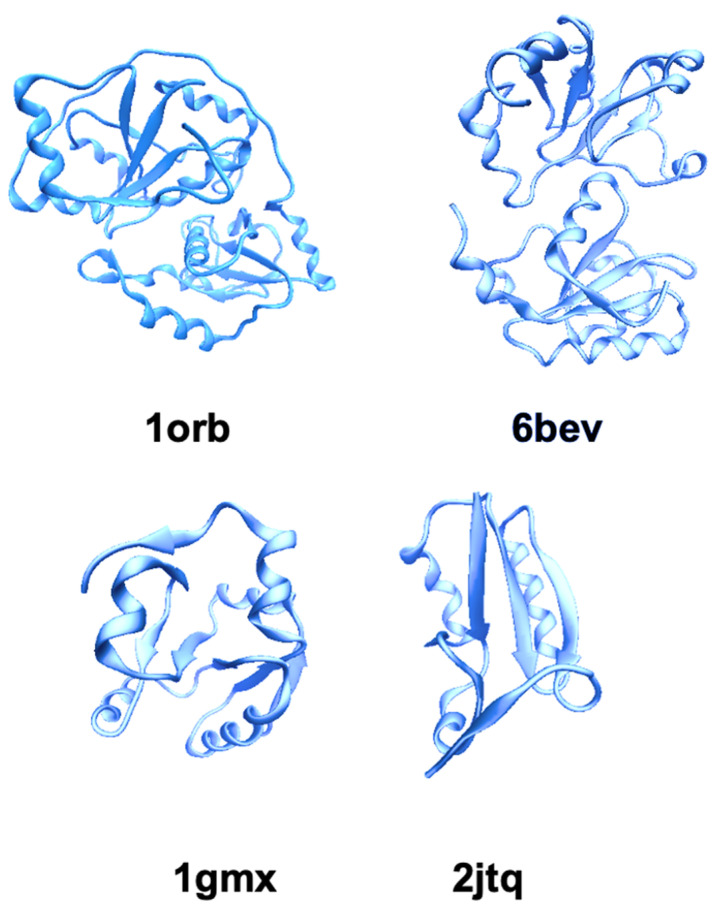
Cartoon representation of 3D structures of thiosulfate: cyanide sulfurtransferases with tandem and single domain. (https://www.rcsb.org/) TSTbov (1 orb) [45], TSTD1 (6 bev) [33], GLlpE (1 gmx) [21] and PspE (2 jtq) [58].

**Figure 3 ijms-23-08452-f003:**
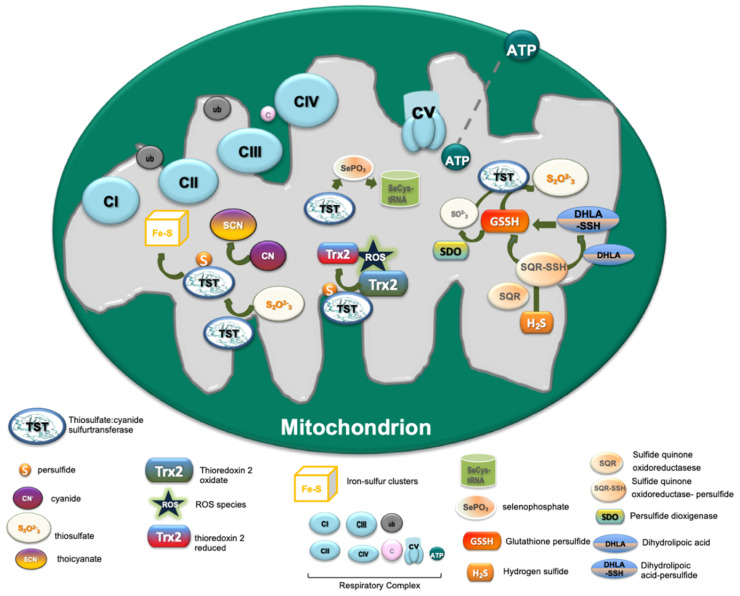
Schematic representation of the functional roles of TST: in cyanide detoxification by thiocyanate formation; in the respiratory complex by Fe-S-cluster formation; in the redox system by thioredoxin and glutathione restoration and hydrogen sulfide oxidation.

**Figure 4 ijms-23-08452-f004:**
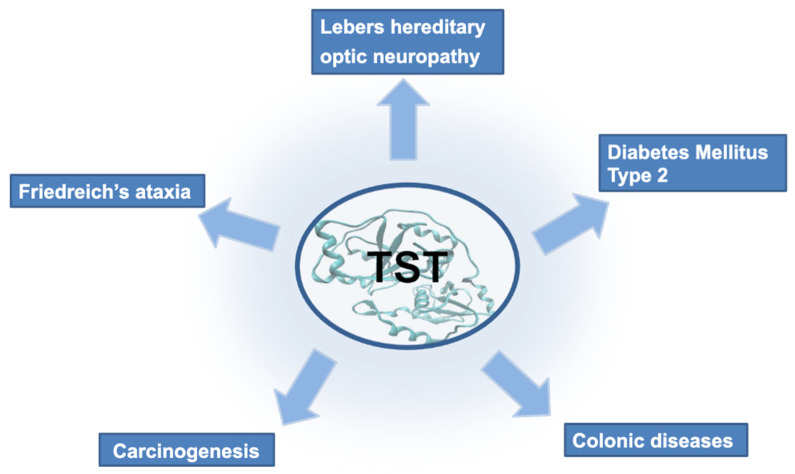
Schematic description of the diseases where TST expression and activity were suggested to play a relevant role.

**Table 1 ijms-23-08452-t001:** Biotechnological applications of the TST enzyme.

TST	Applications	References
*Aureobasidium pullulans*	- Biodegradation of cyanide in cassava wastewater using immobilized rhodanese on alginate	[144]
*Azotobacter vinelandii*	- Cyanide removal from cassava mill wastewater using *Azotobacter vinelandii*; - REPS-hydrogel as enzyme carrier system; - TST-microbubbles addition in PF-hydrogel for the optimization of 3D cellular scaffold	[146,159,188]
*Bacillus pumilus* and *Pseudomonas putida*	- Aerobic cyanide degradation by bacterial isolates from cassava factory wastewater	[160]
*Bos taurus*	- Enzyme therapy in cyanide poisoning: effect of rhodanese and sulfur compounds; - Encapsulation of rhodanese and organic thiosulfonates by mouse erythrocytes; - Characterization of liposomal vesicles encapsulating rhodanese for cyanide antagonism; - Nano-intercalated rhodanese in cyanide antagonism; Reducing cyanide concentrations and enhance fiber digestibility in ruminant animals; - Immobilized rhodanese on sepharose for cyanide detoxification - Rhodanese incorporated in Langmuir–Blodgett films of dimyristoylphosphatidic acid	[147,148,164,168,172,173,174,176,177,178]
*Bos taurus* and *Saccharomyces cerevisiae*	- Biosensor for cyanide detection	[182,183,185]
*Pseudomonas aeruginosa*	- Cyanide detoxification by recombinant bacterial rhodanese; - Involvement of pseudomonas aeruginosa rhodanese in protection from cyanide toxicity	[23,158]
*Rattus norvegicus*	- Immobilized rhodanese on polyacrylamide gel	[163]

## Data Availability

Not applicable.

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
