# Peer review of "Thiosulfate-Cyanide Sulfurtransferase a Mitochondrial Essential Enzyme: From Cell Metabolism to the Biotechnological Applications"

_ijms, 2022, doi:10.3390/ijms23158452_

Round 1

Reviewer 1 Report

Review of the paper entitled “Thiosulfate-Cyanide Sulfurtransferase a mitochondrial essential enzyme: from cell metabolism to the biotechnological applications” by Silvia Buonvino, Ilaria Arciero and Sonia Melino.

      The Authors' paper concerns structural and functional characteristics of thiosulfate: cyanide sulfurtransferase, which is also called rhodanese (EC 2.8.1.1; TST), describing the biological role and biomedical and biotechnological applications of this enzyme. Rhodanese was discovered in the liver rat in 1933 (Lang K., 1933. Die Rhodanbildung im Tierkörper. Biochem Z. 259, 243–256). In 1983, it was located the active site of rhodanese (Hol W. G., Lijk L. J., Kalk K. H., 1983. The high resolution  three-dimensional structure of bovine liver rhodanese. Fund. Appl. Toxicol. 3, 370–376), and three years later, in 1986, the mechanism of transfer of a sulfane sulfur atom from thiosulfate to cyanide. (Horowitz P., Criscimagna N. L., 1986. Low concentrations of guanidinium chloride expose apolar sufraces and cause differential perturbation in catalytic intermediates of rodanese. Biol. Chem. 261, 15652–15658). Recently, the relevance of TST in various diseases has been also highlighted.

This paper is interesting and well written.

 I have only one comment for the Authors to consider.

     In my opinion, it is advisable that the authors extend their review to include information on the formation process of hydropersulfides during mitochondrial H2S oxidation. In this process the main products of H2S oxidation there are thiosulfate and sulfate and enzymes participating in this process include: SQR, ETHE1, sulfite oxidase (SO) and TST. In this process, TST „works” with dihydrolipoic acid (DHLA) that its the reduced form of lipoic acid (LA). The second process in which TST also „works” with DHLA is the generation and transport of sulfane sulfur as well as H2S production from L- and D-cysteine.

     Thus, it seems that the LA/DHLA system is an important player in the sulfur compounds metabolism scene in which TST is involved [Villarejo M, Westley J. Rhodanese-catalyzed reduction of thiosulfate by reduced lipoic acid. J Biol Chem. 1963 Mar;238:1185-6. PMID: 13997286; Cianci M, Gliubich F, Zanotti G, Berni R. Specific interaction of lipoate at the active site of rhodanese. Biochim Biophys Acta. 2000 Aug 31;1481(1):103-8. doi: 10.1016/s0167-4838(00)00114-x. PMID: 11004580; for a review see: Iciek M, Kowalczyk-Pachel D, Bilska-Wilkosz A, Kwiecień I, Górny M, Włodek L. S-sulfhydration as a cellular redox regulation. Biosci Rep. 2015 Nov 25;36(2):e00304. doi: 10.1042/BSR20150147. PMID: 26607972; PMCID: PMC5293568].

     However, the Authors did not dedicate any word to DHLA/LA system in their review paper.

     My decision: minor revision.

Author Response

POINT-TO-POINT response to Reviewer’s Comments

First of all, we would like to thank the reviewers for their constructive and positive comments. We hope that our efforts to address the reviewers’ concerns have succeeded in allowing us to prepare a significantly better version of our manuscript. We have devoted a significant time to follow their useful suggestions and address all the comments point-by-point, as shown below. 
We hope that the new amended manuscript will meet the requirement of the journal and the interest of the scientific community.

Referee 1
Open Review

(x) I would not like to sign my review report
( ) I would like to sign my review report
English language and style
( ) Extensive editing of English language and style required
( ) Moderate English changes required
( ) English language and style are fine/minor spell check required
(x) I don't feel qualified to judge about the English language and style
Is the work a significant contribution to the field?    
Is the work well organized and comprehensively described?    
Is the work scientifically sound and not misleading?    
Are there appropriate and adequate references to related and previous work?    
Is the English used correct and readable?    

Comments and Suggestions for Authors
Review of the paper entitled “Thiosulfate-Cyanide Sulfurtransferase a mitochondrial essential enzyme: from cell metabolism to the biotechnological applications” by Silvia Buonvino, Ilaria Arciero and Sonia Melino.
      The Authors' paper concerns structural and functional characteristics of thiosulfate: cyanide sulfurtransferase, which is also called rhodanese (EC 2.8.1.1; TST), describing the biological role and biomedical and biotechnological applications of this enzyme. Rhodanese was discovered in the liver rat in 1933 (Lang K., 1933. Die Rhodanbildung im Tierkörper. Biochem Z. 259, 243–256). In 1983, it was located the active site of rhodanese (Hol W. G., Lijk L. J., Kalk K. H., 1983. The high resolution  three-dimensional structure of bovine liver rhodanese. Fund. Appl. Toxicol. 3, 370–376), and three years later, in 1986, the mechanism of transfer of a sulfane sulfur atom from thiosulfate to cyanide. (Horowitz P., Criscimagna N. L., 1986. Low concentrations of guanidinium chloride expose apolar sufraces and cause differential perturbation in catalytic intermediates of rodanese. Biol. Chem. 261, 15652–15658). Recently, the relevance of TST in various diseases has been also highlighted.
This paper is interesting and well written.
 I have only one comment for the Authors to consider.
     In my opinion, it is advisable that the authors extend their review to include information on the formation process of hydropersulfides during mitochondrial H2S oxidation. In this process the main products of H2S oxidation there are thiosulfate and sulfate and enzymes participating in this process include: SQR, ETHE1, sulfite oxidase (SO) and TST.
 In this process, TST „works” with dihydrolipoic acid (DHLA) that is the reduced form of lipoic acid (LA).
The second process in which TST also „works” with DHLA is the generation and transport of sulfane sulfur as well as H2S production from L- and D-cysteine.
Thus, it seems that the LA/DHLA system is an important player in the sulfur compounds metabolism scene in which TST is involved [Villarejo M, Westley J. Rhodanese-catalyzed reduction of thiosulfate by reduced lipoic acid. J Biol Chem. 1963 Mar;238:1185-6. PMID: 13997286;
 Cianci M, Gliubich F, Zanotti G, Berni R. Specific interaction of lipoate at the active site of rhodanese. Biochim Biophys Acta. 2000 Aug 31;1481(1):103-8. doi: 10.1016/s0167-4838(00)00114-x. PMID: 11004580; 
for a review see: Iciek M, Kowalczyk-Pachel D, Bilska-Wilkosz A, Kwiecień I, Górny M, Włodek L. S-sulfhydration as a cellular redox regulation. Biosci Rep. 2015 Nov 25;36(2):e00304. doi: 10.1042/BSR20150147. PMID: 26607972; PMCID: PMC5293568].
     However, the Authors did not dedicate any word to DHLA/LA system in their review paper.
     My decision: minor revision.

Answer: We thank the referee for this comment and we agree that the DHLA/LA is an important system in the TST catalysis and it is missing in this review. Therefore, in agreement with the referee comment we changed the manuscript as follows: 
Old version: line 218
However, the detailed molecular role for TST is still debated. TST/Rhodanese catalyzes the transfer of sulfane sulfur from glutathione persulfide (GSSH) to sulfite, which is produced in a reaction catalyzed by persulfide dioxygenase (SDO), i.e. ETHE1, to form thiosulfate [11,93,94]. The ability of the TST to catalyze the production of the H2S [25,75] has suggested the possibility that rhodanese might be a source of H2S in vivo.
New version : line 218
“However, the detailed molecular role for TST is still debated. TST/Rhodanese is a crucial enzyme in the H2S oxidation route that leads to the formation of hydropersulfides (-SSH), thiosulfate and sulfate. TST/Rhodanese catalyzes the transfer of sulfane sulfur from glutathione persulfide (GSSH) to sulfite, which is produced in a reaction catalyzed by persulfide dioxygenase (SDO), i.e. ETHE1, to form thiosulfate [12,95,96]. 
TST is also able to catalyze the formation of H2S using dihydrolipoic acid (DHLA) [97,98] which is also involved in the generation and transport of sulfane sulfur as well as H2S production from L- and D- cysteine [46,98]. The ability of the TST to catalyze the production of the H2S [26,76] has suggested the possibility that rhodanese might be a source of H2S in vivo.”
Old Version 
Figure 3 

New version Figure 3 with the  hydropersulfide DHLA-SSH

We have added the suggested following references
References: 
97 . Villarejo M, Westley J. Rhodanese-catalyzed reduction of thiosulfate by reduced lipoic acid. J Biol Chem. 1963 Mar; 238:1185-6. PMID: 13997286;
46. Cianci M, Gliubich F, Zanotti G, Berni R. Specific interaction of lipoate at the active site of rhodanese. Biochim Biophys Acta. 2000 Aug 31;1481(1):103-8. doi: 10.1016/s0167-4838(00)00114-x. PMID: 11004580; 
98. Iciek M, Kowalczyk-Pachel D, Bilska-Wilkosz A, Kwiecień I, Górny M, Włodek L. S-sulfhydration as a cellular redox regulation. Biosci Rep. 2015 Nov 25;36(2):e00304. doi: 10.1042/BSR20150147. PMID: 26607972; PMCID: PMC5293568

Reviewer 2 Report

This paper summarizes recent advances in TST study, especially its relationship with mitochondrial health and metabolic syndromes. The paper is well organized and the contents are proper. However, I have some suggestions that the authors need to consider before publication.

1. Line 34, convert cyanide in a thiocyanate? should be "to" than "in"?

2. Line 100, aminoacid sequence. This should be another typo error.

3. The authors missed a significant advance in TST related study.  

"Saccharomyces cerevisiae rhodanese RDl2 uses the arg residue of the active-site loop for thiosulfate decomposition. Antioxidants (Basel). 2021 Sep 26;10(10):1525"

"Rhodanese Rdl2 produces reactive sulfur species to protect mitochondria from reactive oxygen species. Free Radic Biol Med. 2021 Dec;177:287-298. "

In the first paper, the authors studied the mechanism of how rhodanese convert ES form to E form, also summarized how other amino acids can help in this process.

In the second paper, the authors studied how rhodanese affect  mitochondrial health in yeast, and found that sulfane sulfur is hydroxy radical quencher.

These two papers represent most recent advance and should be cited.

Author Response

POINT-TO-POINT response to Reviewer’s Comments

First of all, we would like to thank the reviewers for their constructive and positive comments. We hope that our efforts to address the reviewers’ concerns have succeeded in allowing us to prepare a significantly better version of our manuscript. We have devoted a significant time to follow their useful suggestions and address all the comments point-by-point, as shown below.

We hope that the new amended manuscript will meet the requirement of the journal and the interest of the scientific community.

Referee 2

Open Review

English language and style

( ) Extensive editing of English language and style required
( ) Moderate English changes required
( ) English language and style are fine/minor spell check required
(x) I don't feel qualified to judge about the English language and style

Is the work a significant contribution to the field?

Is the work well organized and comprehensively described?

Is the work scientifically sound and not misleading?

Are there appropriate and adequate references to related and previous work?

Is the English used correct and readable?

Comments and Suggestions for Authors

This paper summarizes recent advances in TST study, especially its relationship with mitochondrial health and metabolic syndromes. The paper is well organized and the contents are proper. However, I have some suggestions that the authors need to consider before publication.

  1. Line 34, convert cyanide in a thiocyanate? should be "to" than "in"?
  2. Line 100, aminoacid sequence. This should be another typo error.
  3. The authors missed a significant advance in TST related study.  

 “Saccharomyces cerevisiae rhodanese RDl2 uses the arg residue of the active-site loop for thiosulfate decomposition. Antioxidants (Basel). 2021 Sep 26;10(10):1525"

"Rhodanese Rdl2 produces reactive sulfur species to protect mitochondria from reactive oxygen species. Free Radic Biol Med. 2021 Dec;177:287-298. "

In the first paper, the authors studied the mechanism of how rhodanese convert ES form to E form, also summarized how other amino acids can help in this process.

In the second paper, the authors studied how rhodanese affect  mitochondrial health in yeast, and found that sulfane sulfur is hydroxy radical quencher.

These two papers represent most recent advance and should be cited.

Answer:  We thank the referee for these comments and in agreement we changed the manuscript and added the suggested references:

  1. Saccharomyces cerevisiae rhodanese RDl2 uses the arg residue of the active-site loop for thiosulfate decomposition. Antioxidants (Basel). 2021 Sep 26;10(10):1525"
  2. Rhodanese Rdl2 produces reactive sulfur species to protect mitochondria from reactive oxygen species. Free Radic Biol Med. 2021 Dec;177:287-298.
